# Adjusting Surface Models of Cellular Structures for Making Physical Models Using FDM Technology

**DOI:** 10.3390/polym15051198

**Published:** 2023-02-27

**Authors:** Adrián Vodilka, Martin Koroľ, Marek Kočiško, Jozef Zajac

**Affiliations:** Faculty of Manufacturing Technologies, Technical University of Kosice, 080 01 Presov, Slovakia

**Keywords:** FDM, additive manufacturing, cellular structure, lattice structure, mesh repair, mesh error, polygonization, polygon mesh

## Abstract

In the planning stage of the fabrication process of physical models of cellular structures, a surface model of the structure needs to be adjusted to acquire the requisite properties, but errors emerge frequently at this stage. The main objective of this research was to repair or reduce the impact of deficiencies and errors before the fabrication of physical models. For this purpose, it was necessary to design models of cellular structures with different accuracy settings in PTC Creo and then compare them after the tessellation process using GOM Inspect. Subsequently, it was necessary to locate the errors occurring in the process of preparing models of cellular structures and propose an appropriate method of their repair. It was found that the Medium Accuracy setting is adequate for the fabrication of physical models of cellular structures. Subsequently, it was found that within regions where mesh models merged, duplicate surfaces emerged, and the entire model could be considered as manifesting non-manifold geometry. The manufacturability check showed that in the regions with duplicate surfaces inside the model, the toolpath creation strategy changed, causing local anisotropy within 40% of the fabricated model. A non-manifold mesh was repaired in the proposed manner of correction. A method of smoothing the model’s surface was proposed, reducing the polygon mesh density and the file size. The findings and proposed methods of designing cellular models, error repair and smoothing methods of the models can be used to fabricate higher-quality physical models of cellular structures.

## 1. Introduction

The design, development and implementation of cellular structures enable the fabrication of physical models with the required characteristics unachievable before these structures were introduced. Significant advantages of using cellular structures are improved mechanical properties and object resilience whose weight is, at the same time, reduced, and so is the volume of material needed for their production. Conventional manufacturing technologies were practically incapable of producing these cellular structures in the past. Currently, the fabrication of physical cellular structure models is made possible by the rapid development of additive technologies. One of the most frequently used methods of making physical models is fused deposition modeling (FDM), suitable for making cellular structures. In the planning stage of the process of making physical models of cellular structures using FDM technology, a surface model of the structure needs to be designed so that the structure has the requisite properties. Several options, significantly affecting the quality of the models produced, are available for designing and tessellating computer models of cellular structures. At the same time, errors and model deficiencies often emerge at this stage, notably limiting the quality of fabricated physical models. These errors and deficiencies should be eliminated, or their effect should be reduced as much as possible using FDM technology. These deficiencies are often undetected in the fabrication of physical models of cellular structures. In the context of the frequent use of these models of cellular structures, e.g., in tests of samples or in biomedical applications, these deficiencies may cause a local change in the mechanical properties of the model caused by material in-plane anisotropy, thus making them unsuitable for a given application. Moreover, prior to the fabrication, it is necessary to improve the selected surface model’s properties, namely, to smoothen the model and to verify its mesh density to check whether it lies within the tolerance limit, which, at the same time, reduces the size of the stored model file. Additional research needs to be conducted regarding this issue. Previous research on the issue has been extensively focused on the repair of polygon models, the design of surface models of cellular structures, and the selection of process parameters for the fabrication of cellular structures. This research is aimed at providing an overlay of research in these areas to improve the quality of fabricated cellular structures.

In [1], the authors published a work focused on analyzing typical polygon model deficiencies causing their unfitness for use in selected industries. At the same time, it addressed algorithms available for their repair and the improvement of model properties and topologies to increase their usability in independent applications. The work describes typical defects of polygon models, classified by the authors into local deficiencies in terms of their merging, deficiencies of the model’s topology and deficiencies of the model’s geometry. In view of these deficiencies, polygon model repair algorithms were analyzed and presented in a clear overview, with a description of their effect on the model. In [2], the authors published a study which was focused on non-manifold STL models and their repair method resulting in a small loss of the model’s quality. The benefit of the work is its conclusion that volumetric models with insufficient surface quality are well-repairable, unlike zero-thickness surfaces, which must be redesigned into volumetric models to be printable.

In [3], the authors published a detailed analysis of 3D printing, model design in the process planning stage and the adjustment of models for production by additive technology. The benefit of this work is that it provides an overview of extensive research in this field, classifying process planning algorithms and summarizing current challenges to adjustment for and production by additive technologies, in particular, FDM technology.

In [4], the authors published work, where they examined the results of surface tessellation for exporting the CAD model in the STL format so that the model could be subsequently produced by FDM technology. The resulting model with requisite properties was obtained from the CAD tool’s general midpoint sub-division, with the evaluation criterion being the comparison of the STL volume of the model and its CAD volume. The final difference was 0.03793 mm^3^, which is 0.00017% of the CAD model volume.

In [5], the authors analyzed cross-sectional views of models in terms of the surface roughness, waviness profile and Gaussian filter of the structures. The authors made appropriate use of the measurement and comparison of surface roughness within various filaments.

In [6], the authors designed microstructures with polyether ether ketone (PEEK) and its composites to improve the compatibility of implants of cellular structures as porous hip bone implants fabricated by FDM. The improved design eliminates slight imperfections, allowing for a more stable structure.

In [7], the authors studied the viscoelastic properties of complex cellular structure models manufactured with the PolyJet Matrix—PJM additive technology. Three various photo-curable polymer resin types were used, rheological tests in the form of compressive stress relaxation tests were conducted and the impact of the geometric structure shape and material selection on viscoelastic properties, as well as the most favorable geometric variants of the tested cellular structure models, were determined. The most favorable rheological was adopted and its mean parameters were determined, which enables to match both of the printed model materials and their geometry in the future to create a component with a specific rheological response.

In [8], the authors performed fatigue bending tests on polymer matrix composite material models fabricated using the fused filament fabrication (FFF) technique reinforced with continuous Kevlar fibers. The geometry of the specimens was established according to the ASTM D6272-17 standard. Numerical results were validated against experimental tests. An inspection of the failed surfaces revealed the mechanisms of fiber breakage, matrix cracking and matrix porosity for the static tests, whereas for alternative tests fiber tearing, fiber buckling, matrix cracking and matrix porosity.

In [9], the authors examined the influence of the toolpath strategy on the tensile strength of a model fabricated by the fused filament fabrication technique. They described this technique as inherently directional as the material is deposited in a layer-wise manner; thus, the in-plane material cannot reach the isotropy character when performing the tensile test. This causes the strength of the print components to vary based on the different process planning selections, which include a building orientation and toolpath pattern. They proposed an in-plane isotropic toolpath generation strategy, using which, they were able to increase the tensile strength of specimens by at least 20% under the same printing conditions and process parameters. Regarding this research, it can be suggested that the mechanical properties of specimens fabricated by FDM/FFM technology are highly dependent on the process parameters, especially the toolpath strategy, which can generate local anisotropy of the layer. 

In [10], the authors characterized the influence of the toolpath on the mechanical properties of specimens fabricated using the FDM technique. They stated that the toolpath can have a significant influence on FDM part performance, which is significantly less extensively studied. The structure and its toolpath strategy of specimens are highly dependent on its geometry and design. Changes in toolpath generation using the same process parameters will result in a different mechanical property. A change in the toolpath, causing a change in the structure of the printed part, resulted in significant differences in the mechanical properties of specimens. 

In [11], the authors focused on the design, testing the mechanical properties and the creation of porous structure models in CAD and in the STL format. The benefit of this work is the conclusion that the model in the STL format is sufficient for producing the models but insufficient for FEM simulation due to tessellation-generated deficiencies. In the process of tessellation into the STL format, empty surfaces, as well as other deficiencies that do not satisfy the FEM simulation needs, may appear. 

## 2. Materials and Methods

It is necessary to divide the experimental part into several stages. In the first stage, the model of the selected gyroid cellular structure will be designed in the PTC Creo, together with the tessellation and its export into the STL file. In this part, several accuracy settings are available for the model to be adjusted which need to be compared in GOM Inspect in order to choose the accuracy suitable for the stages of the experiment that would follow. In the second stage of the experiment, emergent deficiencies from the process of adjusting the cellular structures model are located and described, and an appropriate method of their repair is proposed. Repair measures that have been applied would be described and the resulting models verified in terms of the properties required of the model and its manufacturability by FDM technologies. In the third stage of the experiment, it was necessary to focus on the surface model quality, aiming at improving its characteristics in order to be utilized for the fabrication of physical models using FDM technology, i.e., smoothing the model’s surface, reducing the polygon mesh density and reducing the STL format file size.

### 2.1. Cellular Structures

Currently, cellular structures are classified according to their mechanical properties. That is why they are needed in several industries, such as mechanical engineering, aviation, medicine, the army, etc. In recent years, research into cellular structures focused on techniques enabling the creation of products with precise porosity and pore size. The advantages of a design containing a cellular structure involve excellent energy absorption and convenient thermal, acoustic and insulation properties. Yet, the greatest benefit lies in the fact that such a structure shows a high degree of strength at minimum material requirements for its production. As to the cellular structure’s classification, another significant utilization in additive manufacturing is the selection of a suitable unit cell. A basic cell is chosen according to the intended use of the given model. Cellular structures can be divided into two categories—foams and lattice structures. These are subsequently divided into two smaller sub-groups. The foams either have an open or a closed cell. Where lattice structures are concerned, these can be divided into 2D and 3D lattice structures [12].

Several industries require materials satisfying the conditions of fatigue resistance, rigidity, strength, etc., whereas other parameters, such as the low weight of the part or the material in question, need to be satisfied, too [13]. More and more industries are seeking materials with exactly these properties. In this regard, it is indeed the abovementioned foams and architectonic porous materials that can offer specifically those properties. In general, porous materials are defined by a porosity greater than 70% and consisting of a mutually intersected porous mesh of rigid material or beams constituting the edges of the pores or surfaces making up the cells [14,15,16]. A good example of the use of cellular structures in the present is the 3D printing of bone scaffolds with hybrid biomaterials [17,18].

Cellular structures can be manufactured from many materials with different properties such as metals, polymers, ceramics and, in many cases, also composite materials. Metallic cellular structures have been known and used in energy-absorbing applications for many years now. However, when produced in conventional manufacturing processes, they offer irregular pore size and inhomogeneous (uneven) properties to such an extent that each specimen differs from another in the same production batch, which greatly reduces its reliability [14,15,16]. Based on [19], a unique structure cannot achieve its goals if it is fabricated using an unsuitable material. Common materials for the fabrication of cellular structures within the scope of the selected scientific publications are Ti-6Al-4V alloy, 316L stainless steel, polymer resin, titanium, thiol-one polymer, ABS, PLA, etc. [19]. Promising materials for the use of cellular structures in medicine are calcium phosphate-type ceramics (cHAp) as biomaterials, especially for uses where they are in contact with bone structures, due to their chemical similarity to human bone [20]. Another promising material with medical applications is polyether ether ketone (PEEK), a polymer with better lignin biocompatibility than other polymers [21].

Seven different additive manufacturing techniques can be used within the fabrication of physical models of cellular structures. The techniques are as follows: binder jetting (BJ), material extrusion (ME), powder bed fusion (PBF), vat photopolymerization (VP), direct energy deposition (DED), material jetting (MJ) and sheet lamination (SL) [22].

### 2.2. Gyroid Structure

A gyroid is a structure discovered by NASA Scientist Alan Schoen in 1970. The characteristic of this unique structure is the absence of straight lines. In fact, it is a triple periodic minimum surface (TPMS). A TPMS is a surface with zero average curvature, and typical for this surface is a minimal local surface which means that each sufficiently small patch taken from the TPMS has the smallest surface among all the patches created under the same boundaries. We could say that the gyroid is a 3D geometry consisting of intersecting 2D curves which create a strong, robust structure [23,24]. The gyroid structure is promising for medical applications using PEEK/cHAP [14,25]. To create the Gyroid structure, we use the following formula:sin(x) cos(y) + sin(y) cos(z) + sin(z) cos(x) = 0. (1)

### 2.3. Fused Deposition Modeling

Rapid prototyping (RP) is a manufacturing technique that enables the creation of physical objects based on information in the form of computer data. Rapid prototyping is based on the system integration of computer-aided design (CAD) and the RP technique of physical models’ creation. RP is useful for creating small series of physical objects that can be used for various purposes. Additive manufacturing, (AM), is an important RP technique [26]. AM includes manufacturing technologies used in creating physical objects based on the technique of the gradual deposition of layers of material on top of each other. AM offers the option of choosing the right alternative from various materials such as polymers, concrete, metals and composites, depending on the requirements. Compared to conventional methods, additive manufacturing processes are especially suitable for use in small series production, due to high material efficiency thanks to zero or minimum waste produced, the greater efficiency of the resources needed and the possibility of manufacturing models of complex shapes. Currently, the fabrication of physical models using AM technologies is restrained by the limited size, inhomogeneous structure and surface defects of the physical models produced, together with the high costs of technologies of requisite quality and the low speed of these models’ fabrication [27].

Fused deposition modeling (FDM), known as 3D printing, is the main AM technique based on material extrusion. As a manner of 3D printing, FDM was patented by Stratasys in 1989 as a technology following up from stereolithography (SLA), developed by Charles Hull in 1986. The introduction of FDM to the larger public was the result of the previous patents’ expiration in 2005. This made it possible to develop open-source projects of 3D printers, the RepRap Movement and the Fab@Home, which spread 3D printing among the public. The first RepRap printer was the Darwin in 2007 and the second one was the Mandel in 2009, which were complicated to assemble and limited as to their functionality. Innovators were improving the available technology, for example, a Czech developer, Jozef Prusa, who in 2010 released his own improved printer, the Prusa Mendel [28].

FDM is based on the process of extruding a thermoplastic polymer through a nozzle within the print head. Thermoplastic material, most often in the form of a filament, is fed by the extruder to the hot end heated up to a specific temperature. Upon entering the hot nozzle, the filament changes into a half-liquid state and is extruded to the printing pad or on a previously printed layer, forming a printed layer in the XY plane. During solidification, the extruded material partially merges with the material extruded before to acquire the required physical properties of the printed part. These planar layers form a physical 3D model [29].

At present, FDM technology machines vary, with many specific designs having specific properties. Fused filament fabrication (FFF) represents a group of 3D printing technologies. This technique of fabricating physical models is similar but differs from FDM based on the given fabrication technique of a particular technology [30]. According to the type of extruder feeding the filament into the print head, we distinguish direct drive and Bowden-type FDM/FFF technologies. The direct drive extruder, abbr. DE, is an integral part of a movable print head holder, so the filament is fed directly into the hot end. In the Bowden extruder, abbr. BE, the feed mechanism and the print head are separated, and the filament is led through a flexible tube. Bowden tubes are most frequently composed of Teflon due to its flexibility, low friction factor and high thermal resistance as high as up to 250 °C [31]. A significant advantage of DE is the low extrusion delay or filament feed between the extruder and the hot end, resulting in higher-quality printed parts. A major disadvantage of DE is the greater weight of the movable print head’s gantry, caused by the weight of the extruder and the driving stepper motor. A notable advantage of BE is the lower weight of the movable gantry, which might reach a higher printing speed while reducing the force applied on slide bearings. Nonetheless, in BE, it is harder to fine-tune the retraction speed of the filament from the nozzle, and some abrasive filaments could deform the Bowden tube over time [32].

Technological FDM/FFF devices can be classified according to the number of print heads or the number of filaments used in the printing process. A regular technological FDM/FFF machine is a single print head. For more complex components printed from several filaments, multi-material devices are used, most often with two print heads [21]. The process of manufacturing a physical 3D model using the FDM/FFF technology starts with preparing a 3D model of the requisite printed part using CAD systems. The 3D model is then sliced into 2D planar layers of specific layer height or slicing tolerance. A slicer enables the easy setting of process parameters and automatically generates printing nozzle paths and other commands for the FDM/FFF technology, which are to be interpreted by their controller board and stepper controllers [27]. The traditional process of planning physical model production by the FDM/FFF technology can be divided into checking the initial model and the requirements it needs to satisfy, choosing the suitable direction and orientation of the model’s production, preparing support structures, slicing and selecting process parameters of the manufacturing technology [3].

### 2.4. Surface Modeling

CAD systems are based on geometric relations between the basic geometric elements—points, edges, straight lines and curves, surfaces and elements of volume [33]. The relations between these elements can be divided into spatial and topological. The definition of spatial relations is based on defining reference elements including points, axes, planes and coordinate systems. The definition of topological relations is based on the use of logical operations [34]. The methods of computer modeling can be divided into wireframe modeling, surface modeling, boundary surface modeling (B-Rep), space decomposition and solid modeling. Mesh modeling is the method of creating models consisting of faces representing an object’s surface. When the object is represented as a mesh model, its surface may be subject to ambiguities [35]. Surface modeling is a method of creating surface models representing an object as vertex, edges, surfaces, and edge conditions which are finite and spatially defined. Surfaces are defined by boundary conditions and the curvature profile of the surface in space. In surface models, intersecting surfaces can be distinguished, unlike in mesh models. Surface models do not include information on the objects’ volumes and weights. B-Rep modeling may be defined as adjusting a certain surface model to create a finite and enclosed topology according to certain rules [36]. Modeling using spatial decomposition is based on an object’s definition utilizing isomorphic cells, which are smaller than the object itself. This method of modeling is usable when creating inputs for numerical simulations, for example, the finite elements method (FEM). Solid modeling is used for creating volumetric models using features for volumetric element creation, most frequently making use of CAD systems. Solid modeling is more user-friendly than surface modeling, and its main area of use is mechanical engineering [33].

Polygon meshes offer an advantage when used in cases where the model’s surface morphology is too complex to be analytically described as surfaces. For this reason, this way of describing objects’ surfaces is used to great extent in reverse engineering [37]. In the conventional process of physical objects’ digitization, 3D surface coordinates are obtained with respect to the coordinate system of a digitization device or a reference coordinate system. These point coordinates in the form of a point cloud with requisite properties according to the digitization technique, the technology used and the digitization setup, are then used in the process of polygonization [38]. According to certain parameter settings, planar surfaces are created in this process, composed of most frequently three polygons, i.e., straight lines connecting adjacent points at a certain distance to create a planar surface model composed of polygon mesh [39]. This process is executed within CAD systems and polygonizing software, the output of which is most often in the Standard Tessellation Language (STL) format [40].

According to the polygon mesh properties, several mesh types are distinguished. In terms of the space in which the mesh is defined, meshes may be 1D, 2D or 3D. In terms of the element’s hierarchy, they can be straight lines, surfaces and volumetric meshes [41]. In terms of the way in which they create topology structures, they may be divided into structured, partially structured, or unstructured [42]. In terms of the elements’ size ratio, they may be divided into isotropic and anisotropic [43]. In terms of their boundary definition, they may be divided into embedded or body-fitted [44]. In terms of how adjacent elements merge, they may be divided into conforming and non-conforming [45]. In terms of the geometric shape of polyhedral elements, they may be divided into triangular, quadrilateral, tetrahedral and other [41].

### 2.5. Surface Models Quality for the Purpose of Their Manufacture by FDM Technology

In [46], correctness criteria are defined applying to polygon meshes in terms of their practical use. The author divided these criteria into geometric and topological correctness. They explained geometric correctness through the requirements expected of polygon meshes, where the mesh should represent the outer surface of a physical 3D model, free from gaps and intersections. They explained topological correctness through requirements expected to preserve topology that would be the same as that of the nominal model. For the purpose of physical model fabrication using FDM technology, geometric correctness is of utmost importance. In [1], the authors analyzed the atypical deficiencies of polygon models causing their unfitness for purpose in selected industrial applications. They defined typical polygon model defects, which they divided into local defects in terms of merging, defects of the model’s topology and defects of the model’s geometry. In terms of merging, local defects include isolated vertexes, distant dangling elements, edge singularity and vertex singularity. In terms of the model’s topology, the defects include topological noise and errors in polygon orientation. In terms of the model’s geometry, the defects include empty surfaces and gaps in the surface, empty polygons, intersecting surfaces, the beveling of the model’s sharp edges and noise based on the initial model data [1].

As in the process of manufacturing physical models by FDM/FFF technology only volumetric models or surface representations of volumetric models can be used as input, requirements are demanded from model repair methods to create manifold volumetric models [3,47]. In the process of manufacturing physical models by FDM/FFF technology, nominal volumetric CAD models are tessellated in the process of tessellation [3]. The STL format is commonly used as the output from the CAD model tessellation process. Tessellation is the process in which the CAD is approximated and calculated as the mesh model [48].

Non-manifold geometries are virtual geometric shapes that cannot be used to describe physical models in the real world [49]. Non-manifold geometries cannot be unfolded into a planar shape with the normal vectors of its surfaces pointing in the same direction [50]. Non-manifold geometries often emerge as an undesirable phenomenon in the process of CAD models’ tessellation and in the process of polygonization of the point cloud. In the manifold geometry, a particular edge of a manifold mesh connects with only two vertexes, and the particular geometry vertex is defined only as one within the topology of the model. In non-manifold geometry, inner and outer surface orientation cannot be defined with certainty, and this uncertainty is likely to cause an error in the process of physical model fabrication using FDM technology [51]. The emergence of non-manifold geometries may be described by the most common causes, such as the occurrence of a manifold edge intersected with several models’ topologies, the occurrence of a geometry edge vertex defined within several models’ topologies, the occurrence of edges and vertexes remote from the rest of the model, the occurrence of non-manifold objects without the defined thickness of their walls composed of 2D model surfaces, the occurrence of inner surfaces embedded in manifold models, and the occurrence of the phenomenon when a model is defined in a 2D plane without defined thickness and the intersected surfaces are oriented in opposite direction [52].

A model’s fidelity is the degree of exactness of the manufactured physical model compared to the original nominal model. It is difficult to manufacture physical models using additive manufacturing techniques so that the resulting model has sufficient fidelity to the nominal model [3]. A model’s fidelity analysis may be divided into analyzing the fidelity after tessellation and analyzing the fidelity in the model’s production preparation process [11]. While analyzing the model’s fidelity after tessellation, it is advised to conduct an analysis of deviations in the distance of the tessellated surface from the nominal CAD model and compare the difference in the tessellated model volume and that of the nominal CAD model [4]. In checking the model’s fidelity in its production preparation process, it is advised to focus on the emergent staircase effect and on checking the resulting surface quality of the model produced [53].

The accurate prediction of mechanical properties is crucial to the proper application of fabricated parts using the FDM technique. The structure of the part is determined by the toolpath which can be highly dependent on the geometry of a design. Various process parameters cause variations in the mechanical properties within fabricated parts. Changes in the toolpath generation using the same process parameters will result in a different mechanical property [10].

## 3. Results and Discussion

In the beginning, using the Extrude function, a prism model was created with the following dimensions: 60 × 20 × 40 mm. In the current experiment, the selected lattice structure type was Formula Driven, and as for the Cell Type, we chose Gyroid for creating separate structures. Since in the experiment, we do not create the filling of the existing components using lattice structures, having selected the Lattice feature, it was necessary to tick the option Replace body with lattice. Cell wall thickness was selected at 1 mm. The cell size was chosen as 10 × 10 × 10 mm for all structures created during the experiment. 

### 3.1. Accuracy Comparison in Lattice Feature

While creating cellular structures using the Lattice feature in PTC Creo 9.0, it is possible to select the desired accuracy settings of the model designed, choosing from Very Low all the way up to Very High. Choosing a higher accuracy increases the model’s computational load for the computer and increases the size of the model saved in the Part format. It was necessary to choose an accuracy suitable for the subsequent stages of the experiment and check the correctness of the option selected. For this reason, models of different accuracies under the Lattice feature were gradually created, from the lowest to the highest. The created models, shown in Figure 1, were tessellated through Save As and by choosing the exported models’ STL format. In Export STL, parameters were selected, so that the tessellated surface is described as precisely as possible. For this reason, the polygon size parameter of Chord height was selected at its lowest value available, namely, 0.0012. Parameter Angle control, enabling an accurate description of the surface with varying curvature, was selected at its highest value of 1. Identical parameters were used in later stages of the experiment in the process of the tessellation of the created cellular structures models, with adjustments of the polygon mesh within GOM Inspect.

Tessellated models shown in Figure 2 of various accuracies were imported in the STL format in GOM Inspect. A volumetric CAD model could not be imported to be used within the comparison as nominal, because PTC Creo does not enable the creation of a volumetric cellular structure model using the Lattice feature, only its surface representation. In order to compare models of various accuracies, a model with the highest fidelity with respect to whether the ideal geometric model of Gyroid had to be chosen, which would serve as the nominal model in model comparisons, and the Actual Mesh to CAD data option was used as a comparison feature. That is why the Very High model was selected, with the highest fidelity assumed. After that, the deviations in the distance of the tessellated models with respect to the nominal model were compared. To be used in the comparison, a common range of Legend ±0.1 mm was chosen, which is approximately six times the standard sigma (6σ) deviation in the models’ surface comparison with Very Low Accuracy. A comparison of tessellated cellular structure models of various accuracies with respect to the nominal CAD model is shown in Figure 3.

By comparing models of various accuracies, a change in the tessellated models’ parameters could be observed in the STL format. As shown in Table 1, the file size at Very Low and Low was significantly smaller than the other options, which was caused by the low fidelity of the models’ surfaces compared to that of the theoretically ideal geometric model of the Gyroid before tessellation. This could be noticed through the higher standard deviation σ of the surface and, at the same time, through a visual inspection of the models created at Very Low and Low, the geometry of which did not include all parts of the geometry surface, i.e., two areas separated in the edge sections of the model. For this reason, it can be stated that to satisfy the needs of cellular structures fabrication with sufficient surface quality and model fidelity, these models are not suitable. The model at Medium at identical σ had a lower deviation range and smaller STL file size than the model of High Accuracy. Upon a visual inspection of the models shown in Figure 4, it was found that within the tessellation of the models at High and Very High, significant surface deformations of the models’ sharp edges started emerging. It may be assumed that these edge deformations emerged in the tessellation process due to the high fidelity of the complex shaped areas of the models’ surfaces and their detailed descriptions. The existence of these deformed edges is not desirable in the process of cellular structures manufacturing using FDM technology. Upon the selection of Medium Accuracy, these deformed edges could hardly be observed. The visual inspection of the model’s surface quality showed surface quality and model fidelity sufficient to satisfy the needs of cellular structures manufactured using FDM technology. In view of these findings, the subsequent stages of the experiment used cellular structures created under the Lattice feature with Medium Accuracy.

### 3.2. Repair of Mesh Model Defects 

To produce cellular structures using FDM technology, the tessellated model needs to be used as an input for creating nozzle paths in the slicer [3]. This tessellated model can be created to meet the model’s quality and fidelity requirements, or it is possible to use a model that has already been created before. The disadvantage of using a model that has already been created is only a limited possibility of repairing its deficiencies and the model’s geometry errors, as well as a limited possibility of increasing the model’s fidelity. Yet, these models are used to a substantial extent, and there is a demand to take steps to improve the properties of these models. In the process of the models’ tessellation, it is important to check for emergent non-manifold geometries, deficiencies and polygon mesh errors, and to consider the model’s file size, surface quality and fidelity.

At the beginning of the experiment stage, we focused on repairing the polygon mesh deficiencies. Already, the created cellular structure of a Gyroid model from the previous stage was used in PTC Creo 9.0 using the Lattice feature, preserving its original size, with the accuracy selected as Medium. A more detailed visual inspection found discontinuity in the created model, with two small volumetric meshes emerging on the opposite edges of the main volumetric mesh model. For this reason, it cannot be stated that the created model is a single watertight manifold geometry. The Move Geometry feature was used within the experiment, through which the model was again created and moved in the +Y direction by 20 mm. As the structure’s basic cell size was 10 × 10 × 10 mm, the created Gyroid structure model with the size of 60 × 40 × 20 mm merged together smoothly. The model created in this way was tessellated into the STL format with the use of previous process parameters. It was subsequently imported into GOM Inspect. An inspection of this cellular structure found that the created model consisted of six independent geometric models shown in Figure 5a. For illustration purposes, these geometries were rendered in different colors. At the same time, the section view across the YZ plane with an offset of 20 mm in direction of the +X axis in the areas of merging geometric parts shows duplicate surfaces inside the model shown in Figure 5b. For this reason, the tessellated model cannot be considered a manifold geometry.

To check the manufacturability of the cellular structure’s physical model with duplicate surfaces inside the model, this model was imported into the slicer software Cura 5.2.1. The model was placed on a virtual heat bed in the working area as shown in Figure 6a, with the use of machine settings for the custom-made FDM machine. Based on [54], considering the good manufacturability of the gyroid structure by FDM technology, supportless printing was chosen. The process parameters were set for printing the PLA material. The process parameters selected in Cura were the same throughout the experiment.

The selected process parameters include:Single extruder with a nozzle diameter of 0.4 mm;Layer height of 0.12 mm;Line width of 0.4 mm;Infill density of 100%;Printing temperature of 218 °C;Build plate temperature of 60 °C;Print speed of 50 mm/s;Travel speed of 60mm/s;No support generated.

The model was sliced into layers, shown in Figure 6b, and paths were generated to manufacture it using FDM technology in the Gcode format. A visual inspection of the generated toolpaths checked in Cura showed that the separated parts on the model’s geometry were printed into free space, which is why it is necessary to remove these geometries before. In addition, it was observed that in the regions with duplicate surfaces, the toolpath generation strategy inside the model changes, with the emergence of separate walls causing local anisotropy within the layers. These walls emerged across the entire model, within 133 out of 332 layers (40% of layers), as shown in Figure 6c, and it may be assumed that they have a notable impact on the mechanical properties of the cellular structure created [55,56]. It may remain unnoticed in the fabrication of physical models of cellular structures. In the context of the frequent use of these models of cellular structures, e.g., in dynamic simulations of samples or in biomedical applications, the mentioned change of the mechanical properties might be considered, making them unsuitable for a given application.

The first method of repairing the polygon model of this cellular structure was using MeshLab [57]. The model was imported into the system, and then the functions Remove duplicate surfaces, Remove duplicate vertices and Remove non-manifold edges were applied. The resulting model of this particular cellular structure after the repair process of its geometry through MeshLab functions continued to include inner intersecting surfaces; it was not sufficient, and the non-manifold geometry was not repaired.

The second method of repairing the tessellated cellular structure model was using 3D Builder by Microsoft [58]. The 3D builder software tool is available in the operating system Microsoft Windows 10. When the model was imported into GOM Inspect, the six separate geometries were selected with the Select Path function. These models were then separately exported as files in the STL format, preserving their placements with respect to the WCS1 coordinate system when saving them. In the next common step, these separate models were imported into 3D Builder. Once imported, the models were selected and merged together via the Merge function. This function enables merging several geometric models into one, with only the outer geometry of the models preserved. When the models were merged, the resulting model was exported in the STL format and re-imported into GOM, as shown in Figure 7a. An examination of this model showed its homogeneity and the fact that it did not include individual geometries separated from the main geometry part. The section view of the model showed the partial removal of duplicate surfaces inside the model, as shown in Figure 7b. This repair method of non-manifold geometry was unsuccessful, and the resulting model is considered insufficient.

The production of a model resulting from the process of duplicate inner surface repair was verified in Cura. As shown in Figure 8a–c, the generated toolpath visually showed that in the surfaces under repair, the production strategy did not change, and neither did any walls emerge.

At this moment, it could be assumed that the only partial removal of the inner surfaces was due to the fact that these surfaces were only intersected and were normally oriented with respect to each other, whereas the surfaces were not self-intersected to some degree. For Merge to work correctly in 3D Builder, it was necessary to achieve the self-intersection of the geometries. PTC Creo 9.0 was used to ensure the geometries self-intersected, together with the cellular structure model created in the step preceding the tessellation. The prism serving as a reference for creating the first part of the cellular structure was modified and enlarged by +0.002 mm in the +Y direction as shown in Figure 9. In preparing a model to be manufactured by FDM technology, the distance of 0.002 mm is nonsignificant. Through this, the intersection is achieved inside the geometry, as shown in Figure 10, and the outer model geometry did not change in this step, i.e., this step did not influence the model’s fidelity. As in the previous stage, the model was then tessellated into the STL format using the same parameters and imported into GOM Inspect. Separate model geometries were divided into layers and exported into the STL format, preserving their position with respect to the WCS1. These models were subsequently imported into 3D builder and merged. The resulting model was exported and checked in GOM Inspect. When this model was inspected in the YZ plane section view, it showed that due to the surfaces intersecting by 0.002 mm, the polygon model no longer included inner surfaces, and it can be concluded that the non-manifold geometry was repaired.

The production of the model resulting from the repair process of self-intersected surfaces inside the model was checked in Cura. As shown in Figure 11, the generated toolpath showed that within the repaired surface, the production strategy did not change, and neither did any walls emerge. The generated toolpaths are compared in Figure 12. 

### 3.3. Recalculation of the Polygon Model within Required Tolerance

A visual inspection of the model showed the unevenness of the freeform surfaces, the exceedingly sharp angles of the mesh polygons and their high complexity as a structure, not necessary for the given application. For this reason, the model’s polygon mesh was recalculated with the aim of reducing its file size and smoothening out the complex-shaped surfaces without a loss of fidelity to the original model. The first function applied was Thin Mesh, shown in Figure 13, to recalculate the polygon mesh with the aim of reducing the number of polygon edges and vertices and executing this process within the given tolerance resulting in smoothing the model’s complex-shaped surfaces without a significant drop in the quality of the model. The Surface Tolerance parameter was empirically selected at 0.05 mm. A mesh adjusted in this way still contains partially sharp mesh polygons. That is why the Relax Mesh function, shown in Figure 13b, was used afterwards, recalculating the mesh, aiming at creating as even a structure as possible. The number of iterations was selected empirically at five, and the Semi-regular option was disabled, which resulted in deformations to the model’s edges. The STL file size of the resulting model, shown in Figure 14, was 20,386 kB, which is an 81.5% reduction compared to the model prior to recalculation, the size of which was 110,220 kB.

As seen in Figure 15, the manufacturability of the recalculated cellular structure model was checked in Cura. A visual inspection of the toolpath did not display any irregularities, and it can be concluded that the model is suitable for the manufacture of a physical model with FDM technology.

## 4. Conclusions

Through the comparison and visual inspection of the models, it was found that the models with Very Low and Low accuracy had low-quality surfaces and low fidelity to a like-ideal geometric model of the structure, which is why they are not suitable for physical model production. The Medium model at the same σ had a lower Deviation Range and lower STL file size than the High accuracy model. A visual inspection found that when the High and Very High accuracy models were tessellated, significant surface deformations of the model’s sharp edges emerged. At Medium accuracy, these deformed edges could hardly be observed anymore, and a visual inspection of the model’s surface quality and fidelity was sufficient for producing these cellular structures using FDM technology.

In the regions where the model’s geometry was merged, duplicate surfaces were emerging, and the entire model was considered a non-manifold geometry. To check its manufacturability, this model was imported into Cura, showing that within the duplicate surface regions inside the model, the toolpath creation strategy changed and separate walls emerged across the entire model, causing local anisotropy within 40% of layers. Local anisotropy caused by the change in the toolpath creation strategy may affect the mechanical properties of the physical model of the structure produced. The model was repaired in 3D Builder using the Merge function. When the separate model parts were merged, duplicate surfaces inside the model were partially removed, and when its manufacturability was checked in Cura, no significant change in the strategy of the toolpath generation was noted. To completely remove the model’s inner surfaces, the self-intersection of these model parts had to be conducted by 0.002 mm prior to the model’s tessellation. When the model was merged in 3D Builder, its inner walls were completely removed. 

The mesh was recalculated to a 0.05 mm tolerance. The mesh adjusted in this way shows partially sharp mesh polygons, which is why the second function, Relax Mesh, was applied next. The STL file size of the resulting model was 20,386 kB, which indicates an 81.5% reduction compared to the original model prior to recalculation, 110,220 kB. The manufacturability of the recalculated cellular structure model was checked in Cura. The visual control of the toolpath did not show any deficiencies, and it can be concluded that the model is suitable for producing a physical model utilizing FDM technology.

The findings from this research may be subsequently utilized within various applications of cellular structures in engineering. The methods can be used to fabricate higher-quality cellular structures. Typical examples are simulations of samples of cellular structures. This research is important for researchers who produce physical models of cellular structures by custom methods, without considering the verification of the quality of the designed models and its impact on their production. Different CAD software and slicers design models of cellular structures of varying quality, whereas the settings within the design affect the final model.

## Figures and Tables

**Figure 1 polymers-15-01198-f001:**
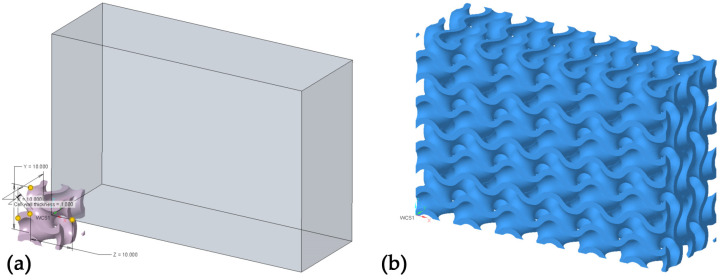
Designing the Gyroid structure in PTC Creo. (**a**) Graphic rendering during the creation of the Gyroid structure through the Lattice feature; (**b**) created Gyroid structure.

**Figure 2 polymers-15-01198-f002:**
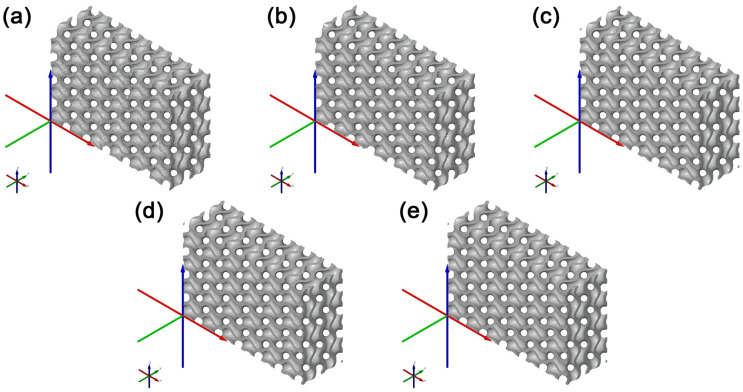
Tessellated models of cellular structures of different accuracies captured in GOM Inspect. (**a**) model Very Low, (**b**) model Low, (**c**) model Medium, (**d**) model High, and (**e**) model Very High.

**Figure 3 polymers-15-01198-f003:**
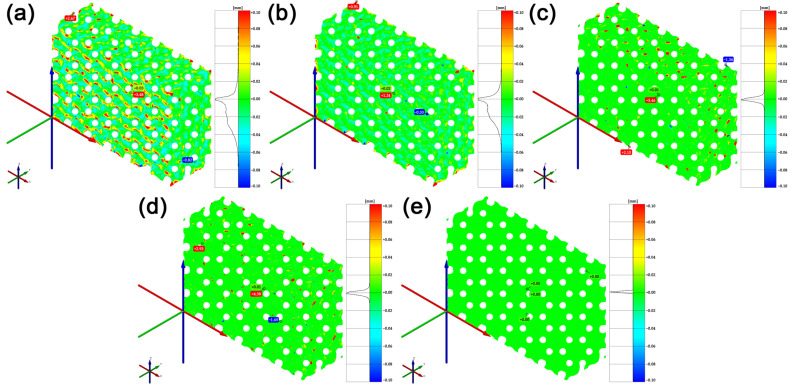
Comparison of tessellated cellular structures models of various accuracies with the nominal CAD model in GOM Inspect. (**a**) model Very Low, (**b**) model Low, (**c**) model Medium, (**d**) model High, and (**e**) model Very High.

**Figure 4 polymers-15-01198-f004:**
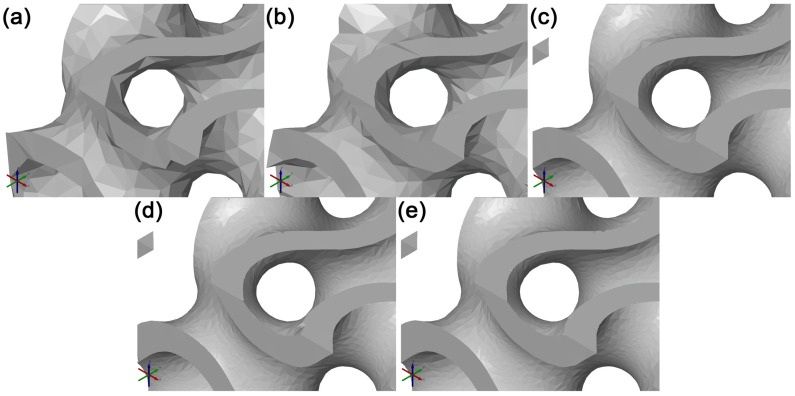
Illustration of the different surface qualities of tessellated cellular structure models of different accuracies, focusing on the boundary edges. (**a**) Model Very Low, (**b**) model Low, (**c**) model Medium, (**d**) model High, and (**e**) model Very High.

**Figure 5 polymers-15-01198-f005:**
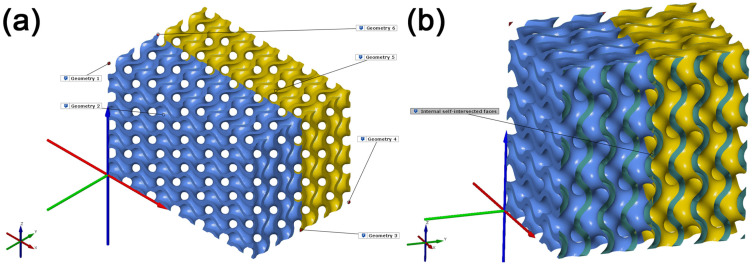
Tessellated cellular structure model composed of six geometric models. (**a**) illustration of separate models making up a single mesh; (**b**) illustration of duplicate areas in the region where the geometric models merge.

**Figure 6 polymers-15-01198-f006:**
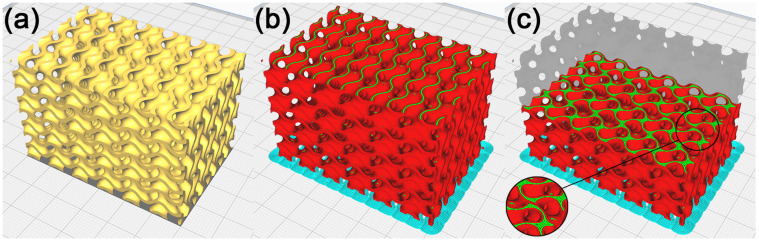
Cellular structure model with duplicated surfaces inside the model. (**a**) Model placed in Cura, (**b**) illustration of the toolpath process in production by FDM technology, and (**c**) illustration of the toolpath in the model’s inner layer.

**Figure 7 polymers-15-01198-f007:**
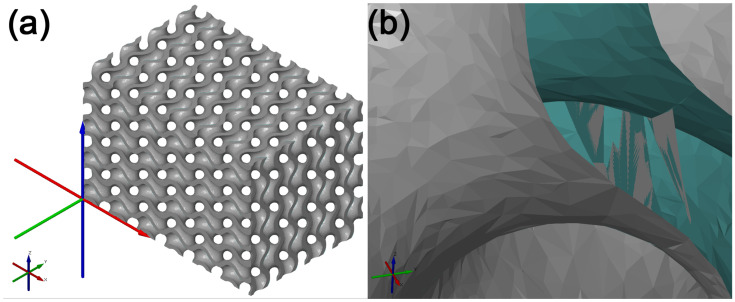
Model resulting from the application of the Merge function. (**a**) View in GOM Inspect; (**b**) section view where the model’s inner surfaces were partially removed in areas of merging geometries.

**Figure 8 polymers-15-01198-f008:**
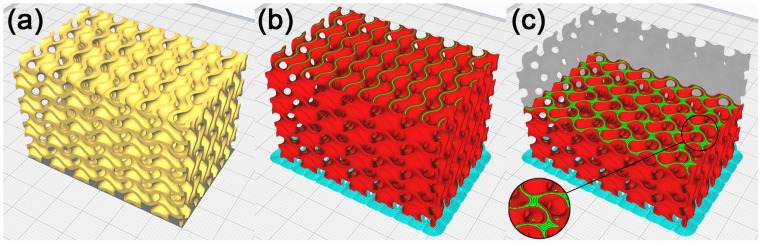
Cellular structure model with partially removed duplicate surfaces inside the model. (**a**) Model placed in Cura, (**b**) illustration of the toolpath process in production by FDM technology, and (**c**) illustration of the toolpath in the model’s inner layer.

**Figure 9 polymers-15-01198-f009:**
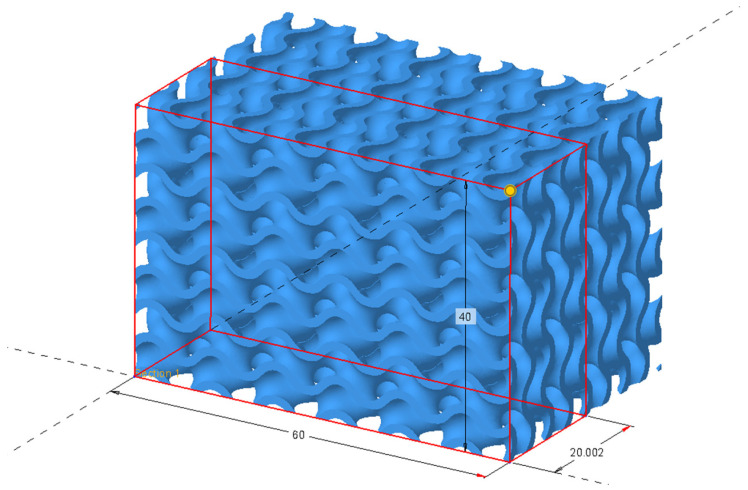
Illustration of the repaired boundary of the basic cellular structure in PTC Creo 9.0.

**Figure 10 polymers-15-01198-f010:**
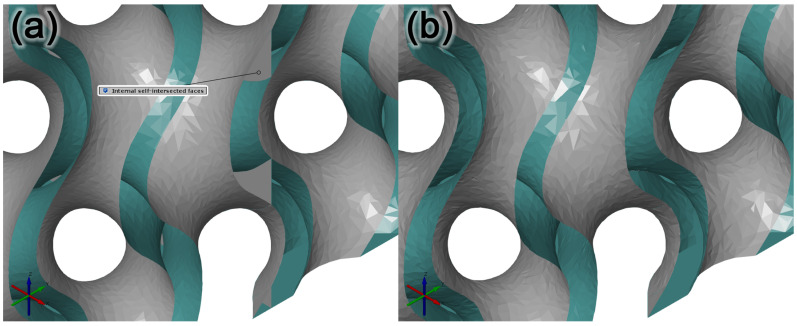
Model with the non-manifold geometry repaired. (**a**) Illustration of self-intersected surfaces inside the cellular structure model; (**b**) illustration of how deficiency was repaired.

**Figure 11 polymers-15-01198-f011:**
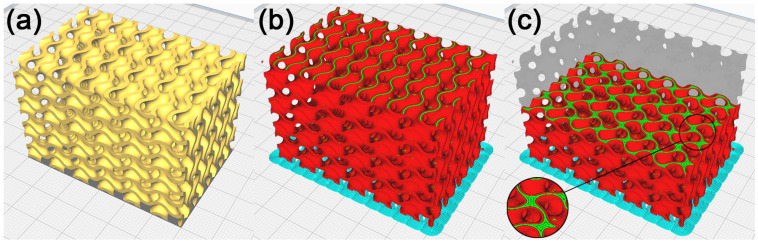
Cellular structure model with removed self-intersected surfaces inside the model. (**a**) Model in Cura, (**b**) illustration of the toolpath process of production by FDM technology, and (**c**) illustration of the toolpath in the model’s inner layer.

**Figure 12 polymers-15-01198-f012:**
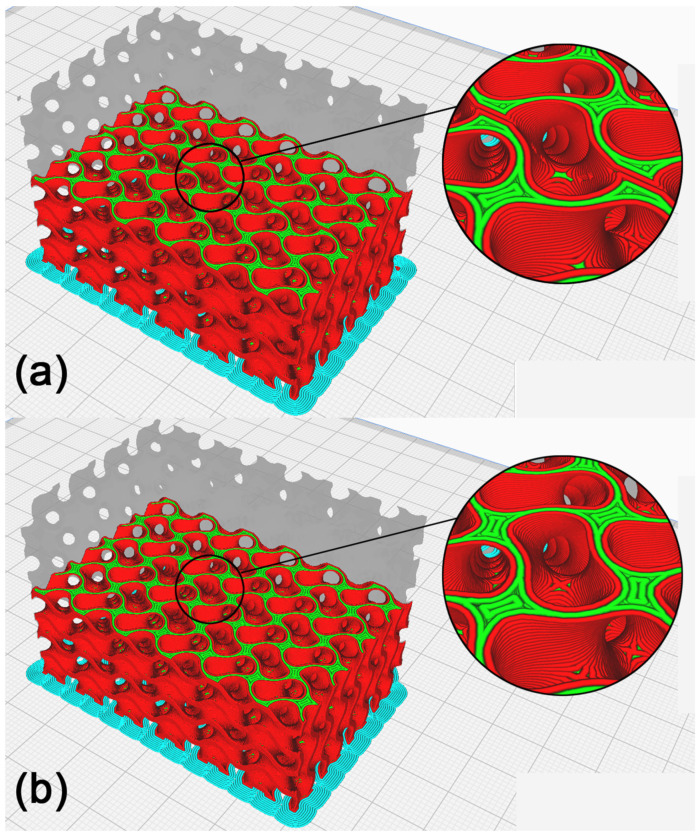
Cellular structure model comparison. (**a**) Illustration of the toolpath process of production before applying the repair method; (**b**) illustration of the toolpath process after repair.

**Figure 13 polymers-15-01198-f013:**
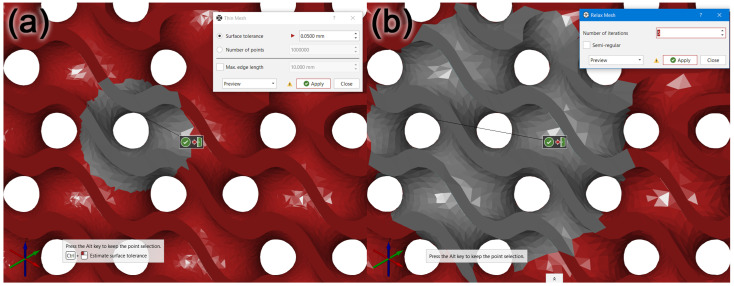
Surface quality in the surface recalculation process. (**a**) Recalculated model using Thin Mesh; (**b**) subsequent recalculation of the model’s area using Relax Mesh.

**Figure 14 polymers-15-01198-f014:**
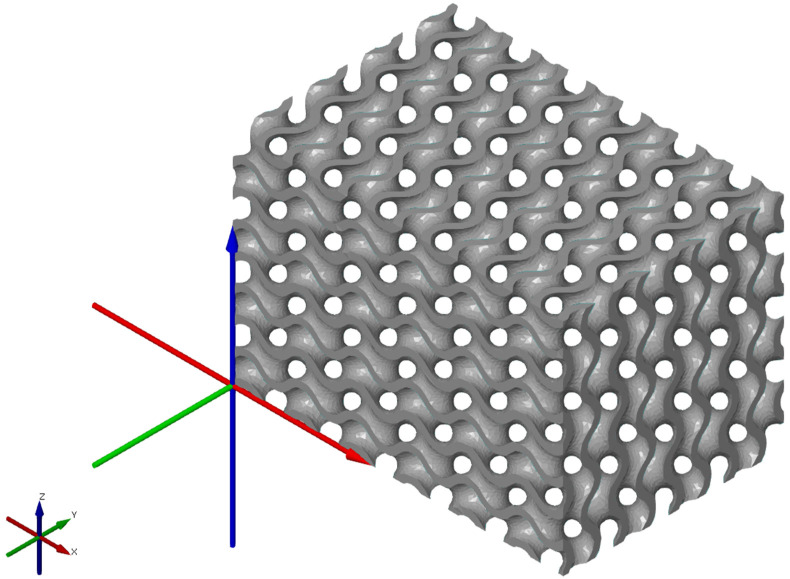
The resulting cellular structure model suitable for the preparation of the physical model manufacturing process using FDM technology.

**Figure 15 polymers-15-01198-f015:**
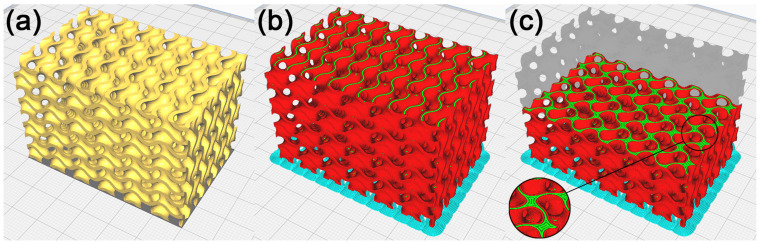
Recalculated cellular structure model. (**a**) Model in Cura, (**b**) illustration of the toolpath process of manufacture with FDM technology, and (**c**) illustration of the toolpath in the model’s inner layer.

**Table 1 polymers-15-01198-t001:** Table of values obtained from the comparison of cellular structure models of various accuracies and the nominal CAD model in GOM Inspect.

**Accuracy**	**Very Low**	**Low**	**Medium**	**High**	**Very High**
STL file size [kB]	4856	6145	55,416	66,524	70,477
Number of triangles	99,434	125,840	1,134,910	1,362,396	1,443,086
Minimum deviation [mm]	−0.82	−0.33	−1.36	−1.65	0
Maximum deviation [mm]	2.67	0.91	2.10	2.93	0
Deviation Range [mm]	3.48	1.24	3.46	4.58	0
Sigma [mm]	0.03	0.02	0.01	0.01	0

## Data Availability

Not applicable.

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
