# Peer review of "Adjusting Surface Models of Cellular Structures for Making Physical Models Using FDM Technology"

_polymers, 2023, doi:10.3390/polym15051198_

Round 1

Reviewer 1 Report

Dear Authors,

The article presents very interesting research results. Below are the comments of the article.

1. The literature review proposes to be enriched with recent publications on cellular structures produced by 3D printing as thin-walled elements. recommends including the following publications in the introduction:

- Viscoelastic Properties of Cell Structures Manufactured Using a Photo-Curable Additive Technology-PJM, DOI10.3390/polym13111895

- A Comparative Study of the Mechanical Properties of FDM 3D Prints Made of PLA and Carbon Fiber-Reinforced PLA for Thin-Walled Applications, DOI10.3390/ma14227062 It is worth referring in the introduction to the construction of cellular structures, largely as thin-walled structures.

2. In point 2.3, it is worth considering the use of the FFF or FDM/FFF vocabulary, because depending on the patents, a certain group of 3D printers represents FDM technologies, and another FFF.

3. It seems that in point 2 Materials - the description could be extended with the properties of materials used in the FDM/FFF technology (perhaps in the form of a table).

4. It seems that it is worth extending the analysis with the supporting material and its impact on the simulation results, because in this technology and in the case of using it to build cellular structures, there may be a problem with cleaning the models. One or two simulation showing support materials might be helpfull but it is not necessary.

5. In table 1, it is proposed to put the number of triangles for STL models, in order to better compare the approximations. It is also worth comparing the level of approximation with the accuracy of a 3D printer in FDM / FFF technology.

It is Minor review and I recommend publishing the article after making the comments mentioned.

Kind regards,

Reviewer

Author Response

Dear Reviewer 1,

on behalf of the authors, we would like to thank you for your insightful and valuable comments and recommendations. We appreciate your review and made efforts to implement your recommendations appropriately within the revision of the manuscript.

1.The literature review proposes to be enriched with recent publications on cellular structures produced by 3D printing as thin-walled elements. recommends including the following publications in the introduction:

- Viscoelastic Properties of Cell Structures Manufactured Using a Photo-Curable Additive Technology-PJM, DOI10.3390/polym13111895

- A Comparative Study of the Mechanical Properties of FDM 3D Prints Made of PLA and Carbon Fiber-Reinforced PLA for Thin-Walled Applications, DOI10.3390/ma14227062 It is worth referring in the introduction to the construction of cellular structures, largely as thin-walled structures.

1.The literature review has been expanded to include the publications you mentioned, which we consider relevant to the research.

2.In point 2.3, it is worth considering the use of the FFF or FDM/FFF vocabulary, because depending on the patents, a certain group of 3D printers represents FDM technologies, and another FFF.

2.Based on your recommendation, the FDM/FFF term was used where considered necessary.

3.It seems that in point 2 Materials - the description could be extended with the properties of materials used in the FDM/FFF technology (perhaps in the form of a table).

3.Based on your recommendation, commonly used materials for the fabrication of cellular structures within the scope of the selected scientific publications were added.

4.It seems that it is worth extending the analysis with the supporting material and its impact on the simulation results, because in this technology and in the case of using it to build cellular structures, there may be a problem with cleaning the models. One or two simulation showing support materials might be helpfull but it is not necessary.

4.Information about supportles printing has been added.

5.In table 1, it is proposed to put the number of triangles for STL models, in order to better compare the approximations. It is also worth comparing the level of approximation with the accuracy of a 3D printer in FDM / FFF technology.

5.We consider this to be a very useful suggestion within the research. The number of triangles has been added to the table.

The authors of this manuscript thank you for the insightful and valuable recommendations, which are useful for current and future work and research.

Kind regards,

Authors.

Reviewer 2 Report

This ‘’Adjusting surface models of cellular structures for making physical models using the FDM technology'' is an interesting topic, The manuscript can be accepted after some major correction
2. More proofreading should be done for typo error
3. Abstract needs to modify: the abstract should contain Objectives, Methods/Analysis, Findings, and Novelty /Improvement. And it too long for abstract
4. The primary defect of this study is that the debate or argument is not clearly stated in the introduction session. Hence, the contribution is weak in this manuscript. I suggest the author enhance your theoretical discussion and arrive at your debate or argument.
5. More work should be done on the literature review and cite more from other on HAp and rGO on PEEK method. It is suggested to add articles entitled to the literature review: https://doi.org/10.1016/j.compositesb.2018.09.065 https://doi.org/10.1016/j.colsurfa.2021.127190
https://doi.org/10.1016/j.rser.2021.111505
https://doi.org/10.1016/j.eurpolymj.2021.110534 https://doi.org/10.1016/j.polymer.2021.123865 https://doi.org/10.1016/j.matchemphys.2021.124485 https://doi.org/10.1016/j.colsurfb.2021.111726 and https://doi.org/10.1007/s42242-020-00098-0 https://doi.org/10.1016/j.ijbiomac.2020.09.175
https://link.springer.com/article/10.1007/s10853-020-04638-y

https://doi.org/10.1016/j.matchemphys.2022.126930

https://doi.org/10.1016/j.colsurfb.2022.112583

https://doi.org/10.1016/j.matchemphys.2022.126454

https://doi.org/10.1016/j.colsurfa.2021.127190
6. More explanations and interpretations must be added for the Results, which are not enough.
7. Please make sure your conclusions section underscores the scientific value-added of your paper and/or the applicability of your findings/results, as indicated previously. Please revise your conclusion part into more detail. It would help if you enhanced your contributions and limitations, underscored your paper's scientific value, and/or the applicability of your findings/results and future study in this session.

Author Response

Dear Reviewer 2,

on behalf of the authors, we would like to thank you for your insightful and valuable comments and recommendations. We appreciate your review and made efforts to implement your recommendations appropriately within the revision of the manuscript.

2.More proofreading should be done for typo error

2.Additional proofreading was carried out to remove typo errors.

3.Abstract needs to modify: the abstract should contain Objectives, Methods/Analysis, Findings, and Novelty /Improvement. And it too long for abstract

3. The abstract has been revised according to your recommendation.

4. The primary defect of this study is that the debate or argument is not clearly stated in the introduction session. Hence, the contribution is weak in this manuscript. I suggest the author enhance your theoretical discussion and arrive at your debate or argument.

4.The importance of the research was appropriately added within your suggestion.

5.More work should be done on the literature review and cite more from other on HAp and rGO on PEEK method. It is suggested to add articles entitled to the literature review: https://doi.org/10.1016/j.compositesb.2018.09.065
 https://doi.org/10.1016/j.colsurfa.2021.127190
https://doi.org/10.1016/j.rser.2021.111505
https://doi.org/10.1016/j.eurpolymj.2021.110534
https://doi.org/10.1016/j.polymer.2021.123865 https://doi.org/10.1016/j.matchemphys.2021.124485 https://doi.org/10.1016/j.colsurfb.2021.111726 and
https://doi.org/10.1007/s42242-020-00098-0
https://doi.org/10.1016/j.ijbiomac.2020.09.175
https://link.springer.com/article/10.1007/s10853-020-04638-y
https://doi.org/10.1016/j.matchemphys.2022.126930
https://doi.org/10.1016/j.colsurfb.2022.112583
https://doi.org/10.1016/j.matchemphys.2022.126454
https://doi.org/10.1016/j.colsurfa.2021.127190

5.Suggested articles have been appropriately added to the literature review and to the text.

6.More explanations and interpretations must be added for the Results, which are not enough.

6.More explenations were added to the results, where considered to be needed.

7.Please make sure your conclusions section underscores the scientific value-added of your paper and/or the applicability of your findings/results, as indicated previously. Please revise your conclusion part into more detail. It would help if you enhanced your contributions and limitations, underscored your paper's scientific value, and/or the applicability of your findings/results and future study in this session.

7.More detail has been added to the conclusion, which has been revised.

The authors of this manuscript thank you for the insightful and valuable recommendations, which are useful for current and future work and research.

Kind regards,

Authors.

Reviewer 3 Report

I don't think the article is relevant to this journal, because it deals exclusively with digital modeling. Additive manufacturing is mentioned only as a potential application, and nothing related to polymers is done.

On the other hand, I encourage authors to continue his work and submit elsewhere. In Additive Manufacturing we are prone to use CAD software and slicers as blackboxes, ignoring that each software has different approaches that might alter the geometrical representation and the final printed model. 

I have included some feedback that might help the authors, in case they choose to submit the manuscript elsewhere:  

Line 25: The final model size was reduced by 25%. It is not clear what they are referring to. Is it the file size? Is it dimensions?

Line 50-52: The following fragment must be edited: "Renown authorities publishing research papers on polygon models and the requirements placed on them are Marco Attene, Tao Ju, Charlie C.L. Wang, Marcel Campen and Leif Kobbelt". This is not a placed to make such calls.

Introduction: Please follow journal practices when referring previous work. There is no need to include author, year and title in the text. That information is already presented in the References.

Sections 2.1 and 2.3 do not belong in Materials and Methods. It is a summary of concepts and basic terms.

Sections 2.4 and 2.5: This does not belong here either. Reconsider what information is actually necessary. For instance, Lines 221-233 doesn't seem necessary: most of the content could be abbreviated.

Section 3: There are some elements that should be moved to "Materials and Methods", such as software, dimension of the models, evaluation criteria, methods (thin mesh, relax mesh, etc). In Results, authors should present exclusively the outcome (diagrams, figures, tables) and its analysis.

Author Response

Dear Reviewer 3,

on behalf of the authors, we would like to thank you for your insightful and valuable comments and recommendations. We appreciate your review and made efforts to implement your recommendations appropriately within the revision of the manuscript.

Line 25: The final model size was reduced by 25%. It is not clear what they are referring to. Is it the file size? Is it dimensions?

The abstract has been revised.

Line 50-52: The following fragment must be edited: "Renown authorities publishing research papers on polygon models and the requirements placed on them are Marco Attene, Tao Ju, Charlie C.L. Wang, Marcel Campen and Leif Kobbelt". This is not a placed to make such calls.

That statement was removed according to your sugestion.

Introduction: Please follow journal practices when referring previous work. There is no need to include author, year and title in the text. That information is already presented in the References.

Introduction was revised according to your suggestions.

Sections 2.1 and 2.3 do not belong in Materials and Methods. It is a summary of concepts and basic terms.

Sections 2.4 and 2.5: This does not belong here either. Reconsider what information is actually necessary. For instance, Lines 221-233 doesn't seem necessary: most of the content could be abbreviated.

It is not clear under which section it is appropriate to place this. These sections have been revised as considered appropriate.

Section 3: There are some elements that should be moved to "Materials and Methods", such as software, dimension of the models, evaluation criteria, methods (thin mesh, relax mesh, etc). In Results, authors should present exclusively the outcome (diagrams, figures, tables) and its analysis.

Section 3 was revised as considered appropriate. Considering the fact that the experiment was divided into three stages and these stages are subsequent to each other, it is difficult to separate Section 3 in a way that the continuousness is preserved.

The authors of this manuscript thank you for the insightful and valuable recommendations, which are useful for current and future work and research.

Kind regards,

Authors.

Reviewer 4 Report

1. In introduction, in line 60, “[1] In 2018, M. Attene [2] published a paper As-exact-as-possible repair of unprintable STL files”. Two references are cited together.

2. Again in Introduction, “In 2018, the authors Ledalla et al. (2018) [4] published their work titled “Performance  Evaluation of Various STL File Mesh Refining Algorithms Applied for FDM-RP Process“, where they examined the results of surface tessellation for exporting the CAD model in  the STL format so that the model could be subsequently produced by the FDM technology”. Such lengthy sentences should be avoided for better readability.

3. In introduction, the authors should mention the motivation behind the present work.

4. Section 2, Materials and Methods , is un-necessarily made lengthy with so many previous reporting’s. If required, it should be highlighted in introduction. As per our observation, this section should focus the material used, and methodology adopted in present work. This section should be reduced.

5. As there is no separate section on discussion of results. The result section should be renamed as Results ad Discussion. Also the reasoning behind the results or observations made in present work should be justified with support of previous reporting’s.

6. The conclusion made are too lengthy, also it should be highlighted in points for better readability.

Author Response

Dear Reviewer 4,

on behalf of the authors, we would like to thank you for your insightful and valuable comments and recommendations. We appreciate your review and made efforts to implement your recommendations appropriately within the revision of the manuscript.

1.In introduction, in line 60, “[1] In 2018, M. Attene [2] published a paper As-exact-as-possible repair of unprintable STL files”. Two references are cited together.

1.References were revised according to your suggestion.

2.Again in Introduction, “In 2018, the authors Ledalla et al. (2018) [4] published their work titled “Performance Evaluation of Various STL File Mesh Refining Algorithms Applied for FDM-RP Process“, where they examined the results of surface tessellation for exporting the CAD model in the STL format so that the model could be subsequently produced by the FDM technology”. Such lengthy sentences should be avoided for better readability.

2.Introduction was revised according to your suggestions.

3.In introduction, the authors should mention the motivation behind the present work.

3.Motivation behind the work was added within the introduction as suggested.

4.Section 2, Materials and Methods , is un-necessarily made lengthy with so many previous reporting’s. If required, it should be highlighted in introduction. As per our observation, this section should focus the material used, and methodology adopted in present work. This section should be reduced.

4.Section 2 was revised according to suggestions from reviewers.

5.As there is no separate section on discussion of results. The result section should be renamed as Results ad Discussion. Also the reasoning behind the results or observations made in present work should be justified with support of previous reporting’s.

5.Section 3 was renamed and revised according to your suggestions.

6.The conclusion made are too lengthy, also it should be highlighted in points for better readability.

6.Conclusion was revised according to your suggestions and suggestions of other reviewers.

The authors of this manuscript thank you for the insightful and valuable recommendations, which are useful for current and future work and research.

Kind regards,

Authors.

Round 2

Reviewer 3 Report

Dear authors,

I still fail to see how a work entirely focused on digital modelling might be relevant to the journal Polymers. I have not received any response from the editor (or the journal) about this matter. Based on this fact, my decision of rejecting the manuscript remains unchanged.

However, because two other reviewers have a different position, I think it's up to the editor to judge on the matter, and maybe proceed to publish with a non-unanimous decision.

I add some comments that might be useful:

Abstract (and Text): extensive editing of style is still required.

Lines 57-61: is there any mention of these issues in the literature? Considering the low accuracy of AM methods, I am not sure that defects in digital models can be differentiated from manufacturing defects. As I mentioned, it is a very interesting issue, but it's really to make a distinction between modeling and manufacturing errors in practice.

In some sections of the manuscript the authors comment that errors in digital model are around 0.002 mm. Considering that the accuracy of the FDM/FFF is about ± 0.5 mm, my intuition is that these modelling errors might not be significant in the process. Furthermore, the simulations run in Cura show no evidence of any effect on the extruder path or part geometry.

Section 2.3: Again, I fail to see the relevance of this summary of FDM history. The work focuses on digital modelling and this section does not add anything to the discussion. No AM is done whatsoever in this work.

Lines 427-468: Origins and commercial information of the software is not relevant. It is much more important what will be specifically done with the software in this work, or what capabilities/restrictions are critical to the task.

Author Response

Dear Reviewer 3,

on behalf of the authors, we would like to thank you for your insightful and valuable comments and recommendations. We respect your decision, however, we will also make a revision based on your comment, so that the editorial office can decide. We appreciate your review and made efforts to implement your recommendations appropriately within the revision of the manuscript.

Abstract (and Text): extensive editing of style is still required.

The abstract has been revised as considered appropriate.

Lines 57-61: is there any mention of these issues in the literature? Considering the low accuracy of AM methods, I am not sure that defects in digital models can be differentiated from manufacturing defects. As I mentioned, it is a very interesting issue, but it's really to make a distinction between modeling and manufacturing errors in practice.

We decided to clarify this statement, adding the literature as context. Also, Figure 12 has been added to point out local anisotropy caused by toolpath strategy changes within 40% of model.

In some sections of the manuscript the authors comment that errors in digital model are around 0.002 mm. Considering that the accuracy of the FDM/FFF is about ± 0.5 mm, my intuition is that these modelling errors might not be significant in the process. Furthermore, the simulations run in Cura show no evidence of any effect on the extruder path or part geometry.

Cellular structure was not enlarged by +0.002 mm, only intersection within separate models was created by 0.002 mm,inside structure, while outer geometry of the structure remained the same.

Section 2.3: Again, I fail to see the relevance of this summary of FDM history. The work focuses on digital modelling and this section does not add anything to the discussion. No AM is done whatsoever in this work.

Given that part of the manuscript deals with the influence of process parameters (toolpath strategy and anisotropic layers) on the quality and mechanical properties of the model produced by FDM and FFM, we consider it appropriate to include a description of FDM as one of the basic techniques of additive manufacturing.

Lines 427-468: Origins and commercial information of the software is not relevant. It is much more important what will be specifically done with the software in this work, or what capabilities/restrictions are critical to the task.

Software descriptions were removed within your suggestions.

The authors of this manuscript thank you for the insightful and valuable recommendations, which are useful for current and future work and research.

Kind regards,

Authors.

Round 3

Reviewer 3 Report

Abstract: The previous version was better. The problem was not related to the content, but style. I recommend getting external assistance to improve the English not just in the abstract but in the text overall. 

Figure 2 is not adding much to the discussion, as no difference is evident between models.

I am satisfied with this version of the document, although it still requires some heavy proofreading. There are grammatical errors in different sections.

Author Response

Dear Reviewer 3,

Abstract: The previous version was better. The problem was not related to the content, but style. I recommend getting external assistance to improve the English not just in the abstract but in the text overall.

The manuscript has been revised by a certified translation company in our home country. We have the proof available. The authors have additionally proofread the manuscript within our capabilities. The Article Processing Charges includes minor English editing by native English speakers by MDPI. The APC does not cover extensive English editing. Paper could be returned at the English editing stage of the publication process if extensive editing is required, we will be in contact with the editor.

Figure 2 is not adding much to the discussion, as no difference is evident between models.

Figure 2 is related to Figure 3, with the differences in the models being more noticeable within the high-resolution figure.

The authors of this manuscript thank you for your recommendations.

Authors.